# Association between Stroke and Abdominal Obesity in the Middle-Aged and Elderly Korean Population: KNHANES Data from 2011–2019

**DOI:** 10.3390/ijerph19106140

**Published:** 2022-05-18

**Authors:** Jong Yeon Kim, Sung Min Cho, Youngmin Yoo, Taesic Lee, Jong Koo Kim

**Affiliations:** 1Department of Neurosurgery, Yonsei University Wonju College of Medicine, Wonju 26426, Korea; jjongse@hanmail.net (J.Y.K.); ns1287@hanmail.net (S.M.C.); 2Department of Medicine, Graduate School, Yonsei University Wonju College of Medicine, Wonju 26426, Korea; youngminyoo@yonsei.ac.kr; 3Department of Family Medicine, Yonsei University Wonju College of Medicine, Wonju 26426, Korea; ddasic123@yonsei.ac.kr; 4The Study of Obesity and Metabolic Syndrome, Korean Academy of Family Medicine, Daejeon 35365, Korea; 5Institute of Global Health Care and Development, Wonju 26426, Korea

**Keywords:** stroke, abdominal obesity, waist circumference, KNHANES, risk factor

## Abstract

Obesity and overweight status are primary risk factors for stroke. A relative small number of studies has analyzed the association of abdominal obesity, a crucial indicator for insulin resistance with stroke, compared to general obesity. We aimed to reveal 31,490 records from the Korea National Health and Nutrition Examination Survey (KNHANES). Logistic regression was used to identify the association of abdominal obesity with the risk of stroke. For the multivariate model, covariates were determined based on the cardio-cerebro vascular prediction models. In the sex-specific multivariate logistic regression analysis (including age, antihypertensive drug, diabetes, current smoking, and systolic blood pressure as confounders), the elevated waist circumference (WC) in women was significantly associated with the increased risk for stroke. In case of the categorized form of WC, we discerned the non-linear relationships between WC and the stroke status. The sex-specific associations between the abdominal obesity and stroke status were shown and their relationship pattern exhibited non-linear relationships.

## 1. Introduction

Stroke has mainly two subtypes (i.e., ischemic and hemorrhagic stroke) and is a disease that exerts a high global burden. In the US alone, stroke events occur in about 795,000 population annually, and the increased prevalence of stroke is estimated to affect 3.4 million people between the years 2010 and 2030 [1,2,3,4]. In Korea, approximately 105,000 patients experience new or recurrent stroke each year, indicating that a new case arises every 5 min [5]. With increasing age, the incidence of stroke escalates as follows: 20 and 3297 per 100,000 person-years in a population aged ≤ 44 AND ≥ 85, respectively [5]. Using the Korean National Health Insurance Claims Database, Lim et al. calculated the cost of stroke to be 3737 billion Korean won in 2005 [6].

Obesity and being overweight have been considered as primary risk factors for stroke [7]. The Nurses’ Health Study reported that both obesity and weight gain in women were related to the increased risk of ischemic and total stroke [8]. A meta-analysis of a total of 33 cohorts including approximately 310,000 subjects showed positive relationships between the continuous levels of waist circumference (WC) and the risk of ischemic stroke and hemorrhagic stroke [9]. A study that enrolled about 21,000 US men reported that each unit increase of BMI was related to a 6% increased risk of total, ischemic, and hemorrhagic stroke [10].

There has been a relatively small number of studies reporting on the relationship between the risk of stroke and abdominal obesity as defined by WC [11], compared to general obesity based on the body mass index (BMI) [12,13]. Suk et al. [11] analyzed subjects enrolled in the Northern Manhattan Stroke Study (NOMASS) and demonstrated that abdominal obesity is an independent and potent risk factor for ischemic stroke. Winter et al. [14] suggested that abdominal adiposity exhibited a graded and significant relationship with the risk of stroke and transient ischemic attacks (TIA) analyzing subjects visited the Departments of Neurology of the Klinikum Mannheim and Klinikum Heidelberg. However, they failed to replicate these findings when analyzing data obtained from the DETECT (Diabetes Cardiovascular Risk-Evaluation: Targets and Essential Data for Commitment of Treatment) study [15]. Cho et al. [16] analyzed 21,749,261 Koreans who underwent the Korean National Health Screening between 2009 and 2012 and suggested a significant linear relationship between WC and the risk of ischemic stroke.

Taken together, several studies that analyzed the association between WC and stroke have been provided, but their findings were limited and inconclusive. Therefore, it is crucial to check the previously reported but inconsistent findings using another dataset. This study aimed to reveal whether the significant relationship between WC and the risk of stroke persisted after adjusting for covariates and if their trend showed a linear or non-linear relationship in another Korean public dataset.

## 2. Materials and Methods

### 2.1. Study Population

To demonstrate the association between abdominal obesity and risk of stroke, we analyzed the 2011–2019 Korea National Health and Nutrition Examination Survey (KNHANES). KNHANES is conducted annually by the Korea Centers for Disease Control and Prevention (K-CDC), and employs a complex, multistage probability sample to obtain nationally representative data [17,18]. All of the participants in the 2011–2019 KNHANES signed an informed consent form. All data were accessed in compliance with the Helsinki Declaration. The individual approval of the Institutional Review Board of Wonju Severance Christian Hospital was waived since the KNHANES data are publicly available and all subjects in these surveys are fully anonymized and un-identified. In detail, the dataset was compiled from the KNHANES official website (https://knhanes.kdca.go.kr/knhanes/eng/index.do, accessed on 1 December 2021) after database access permission was granted.

Kim et al. [19] reported that the prevalence of stroke in overall group was 1.71, besides, that was 0.53 in population aged 19–54 years and was 3.72 in population aged 55–74 years. We enrolled subjects aged ≥ 40 to obtain generalized findings for the Korean population and to minimize the bias of distribution with regard to smoking status between the young and elderly groups. Exclusion was then conducted for participants who had incomplete data on medical history, lifestyle indices, and anthropometric and laboratory indices [i.e., WC, systolic blood pressure (SBP), total cholesterol (TC), and high-density lipoprotein cholesterol (HDL-C)]. Finally, 31,490 participants were included in this study.

### 2.2. Definition of Stroke

Based on self-reported questionnaire-based information, the participants who were previously diagnosed with a cerebral infarct or hemorrhage were categorized under the stroke group. Specifically, the presence of a stroke was defined if “Yes” was marked in one or more of the following items: stroke diagnosis by a doctor; current prevalence of stroke; experience with stroke treatment; and presence or experience of stroke-associated sequelae.

### 2.3. Indices for Abdominal Obesity

There have been several indices used to represent abdominal obesity. Suk et al. [11] implemented the waist-to-hip ratio (WHR) as the index for abdominal obesity. Winter et al. [14] used three indices, WC, WHR, and waist-to-stature ratio (WSR), to reflect abdominal obesity. In the Korean study, the six WC groups categorized by intervals of 5 cm were used as the index. Motivated by these studies, we used three forms of WC, the continuous, binary, and categorized forms as the index for abdominal obesity. According to the Korean Society for the Study of Obesity, abdominal obesity was defined as having a waist circumference ≥ 90 cm for men and ≥85 cm for women [20,21].

### 2.4. Covariates

We determined the risk factors used for the establishment of CVD risk models, including the Framingham Risk Score [22], pooled cohort equations (PCEs) [23], and revised PCEs [24] as covariates. These models used eight predictors, including age, sex, TC, HDL-C, SBP, hypertension medication, current smoking, and diabetes.

Information about hypertension medication was obtained from the self-reported questionnaire. The diabetic group comprised participants who were previously diagnosed with or showed fasting glucose ≥126 mg/dL [25]. Smoking status was categorized into two statuses: current smoking or not as similar with previous studies [22,23,24].

### 2.5. Statistics

All data was reconstructed and preprocessed using R language (version 4.0.1, R Foundation for Statistical Computing, Vienna, Austria) [26]. Statistical analysis was performed using the R language. Continuous variables such as age, SBP, WC, and laboratory values were analyzed using Student’s *t*-test. For categorical variables, the chi-square test was utilized.

To estimate the total population that the data would represent, we employed the sampling weights determined by the data constructors. After adopting the weight values, we analyzed the crude or adjusted odds ratio (OR) of WC for the prevalence of stroke. Logistic regression (LR) analysis was basically performed using following equation: *stroke status* (dependent variable) *~WC* (independent variable) *+ covariates*. A *p*-value of <0.05 was considered to be statistically significant.

## 3. Results

### 3.1. General Characteristics

Sex-specific general characteristics according to stroke status are presented in Table 1. Korean men placed in the stroke group had the following characteristics: old age, higher WC, more abdominal obesity, higher SBP, less current smoking, more anti-hypertensive medication, more diabetes, lower TC, and lower HDL-C, compared to no stroke men (Table 1). In the stroke group for women, the following clinical and laboratory variables exhibited similar patterns with those of the stroke group for men: old age, high WC, considerable abdominal obesity, high SBP, many hypertension medications, diabetes, and low levels of TC and HDL-C.

When analyzing the unweighted subjects, the prevalence of stroke (%) was 3.86 (507/13,129) and 2.48 (456/18,361) in Korean men and women, respectively. With the weighted population, the sex-specific prevalence of stroke was 2.98 in men and 2.17 in women. The trend for the ratio of stroke from 2011 to 2019 is described according to two age groups (Figure 1). In both Korean men and women, the monotonically increased or decreased patterns for the prevalence of stroke did not appear as the observation year changed from 2011 to 2019. In Korean men, the prevalence of stroke in the 40–59 and 60 to 80 age groups was 0.44 to 1.95 and 5.59 to 7.4 (percent), respectively. In Korean women, the ratio of stroke in two groups was 0.58 to 1.19 and 2.77 to 5.76, respectively.

When analyzing the weighted subjects, the prevalence of abdominal obesity (%) were 32.7 and 30.4 in Korean men and women, respectively. In the stroke group, the prevalence of abdominal obesity was 37.4 and 44.9 in men and women (weighted population), respectively. The sex-specific ratio of abdominal obesity in the stroke group from 2011 to 2019 was 31.06 to 47.9 in men and 27.8 to 64.6 in women (Figure 2).

### 3.2. Relationship of the Continuous and Binary form of WC with the Risk of Stroke

Table 2 shows the degree of relationship between the continuous form and binary forms (i.e., presence and absence of abdominal obesity) of WC and the stroke status using univariate and multivariate LR. In Korean men, the continuous level of WC was significantly associated with the risk of stroke in Model 2. However, this positive association changed negative and insignificant relationships in Model 3 and 4, respectively. When using the binary form of WC (cut-off of WC: 90 cm) as an independent variable, the presence of WC was inversely related with the increased risk of stroke in the multivariate model, respectively (Models 2 and 3 in Table 2).

In the case of Korean women, the continuous form of WC (cm) was significantly associated with the increased risk of stroke after adjusting clinical, anthropometric, lifestyle-related, and laboratory variables (Model 3 in Table 2). Also, when using the binary form of WC (cut-off of WC: 85 cm), the presence of abdominal obesity was significantly related to the stroke status in the univariate and age-adjusted models. However, this association yielded insignificant results in other multivariate models (Models 2 and 3 in Table 2).

### 3.3. Relationship of the Five Groups of WC with the Risk of Stroke

We categorized the continuous form of WC into five groups according to the cut-offs obtained from quintile and empirical selection (i.e., six groups at intervals of 5 cm between 75 and 95 in men and between 70 and 90 in women). When setting the quintile method as cut-off for five WC groups, quintile 4 (Q4) among the Korean men exhibited the lowest adjusted OR (aOR) among five groups. The Q4 (Figure 3A) had men with WC ranging from 88.4 to 93.3 cm. In case of empirical cut-offs (i.e., 75, 80, 85, 90, and 95 cm in men), the group having 75 to 79 cm of WC levels in men exhibited the best aOR among five groups (Figure 3C).

Among the Korean women, Q3 and group 3 exhibited the highest aOR among five groups (Figure 3B,D). Q3 included Korean women with WC levels that ranged from 78.2 to 83 cm. In all cases, non-linear relationships between WC and the stroke status were shown.

## 4. Discussion

This study analyzed several forms of WC, including the continuous levels, the binary status via sex-specific WC cut-off [27,28], and five groups made by quintile and empirical cut-offs to reveal the association between abdominal obesity and stroke. In case of the continuous WC level, the high level of WC in Korean women was independently associated with the presence of stroke after adjusting for clinical and laboratory covariates. Besides, the presence of abdominal obesity exhibited an inverse relationship with the risk of stroke in Korean men. When categorizing WC levels into five groups based on the quintile or per 5 cm intervals, a non-linear relationship between WC levels and stroke status was observed. 

Five pathophysiologic components are used to diagnose metabolic syndrome (MetS), among which WC is considered to reflect the status of insulin resistance [29]. The International Diabetes Federation group pinpointed the WC levels as the core mechanism of MetS, and determined it as the obligatory component for MetS [29]. The European Group for the Study of Insulin Resistance (EGIR) implemented 94 cm of WC in men and 80 cm in women for determining MetS [30]. The National Cholesterol Education Program, Adult Treatment Panel III (NCEP-ATP III) provided in 2001 the sex-specific WC cut-offs (102 cm in men and 88 cm in women) for the diagnosis of MetS [31]. However, the distribution of WC levels differed according to ethnicity, and recently, several expert groups have concluded to use the population-specific WC distribution or cut-off [29,32,33]. For determining abdominal obesity in this study, the Korea-specific WC cut-off was implemented [27,28]. However, using the Korean WC cut-off yielded insignificant results. It might be due to the non-linear relationship between the WC level and the risk of stroke. Moreover, Cho et al. [16] suggested the optimal WC levels of 85 cm in Korean men and 78 cm in Korean women to predict the risk of ischemic stroke.

A study analyzed the burden of stroke in Europe and reported that the prevalence of stroke increased annually during 1990 to 2017 [34]. The Global Burden of Diseases, Injuries, and Risk Factors Study (GBD) applied GBD 2019 analytical tools to estimate stroke prevalence and reported that the age-standardized stroke prevalence rate decreased between 1990 and 2019 by 6.0% (95% uncertainty interval: 5.0–7.0) [35]. Moreover, this study noted that substantial between-country variations in the age-standardized stroke prevalence rate materialized [35], specifically the age-standardized stroke prevalence in Southeast Asia, East Asia, and Oceania was reported to fluctuate between 1990 and 2008 and monotonically increase between 2010 and 2019. A study from the Epidemiology Research Council of the Korean Stroke Society reported that across all age groups, the stroke prevalence in men was higher than that in women, while the ratio of stroke in adults aged ≥ 50 years did not manifest any specific secular trends from 1998 to 2014. The stroke prevalence calculated in our study had similar trends with what has been provided by other studies, indicating the generalized results for the Korean adult population.

Recent studies attempted to identify the association between abdominal obesity and risk of stroke and stroke-related mortality [36,37,38]. When using WC for the abdominal obesity index, results obtained were either insignificant or inconsistent with our previous findings [36,37]. These studies used WC adjusted by hip or height for the alternative index for abdominal obesity, and yielded significant association between central obesity and stroke. Cong et al. [38] reported that subjects with high BMI and normal WC did not exhibit increased risk for stroke while participants with abdominal obesity showed an increased risk for stroke, mostly independent with BMI, specifically men. Two main pathophysiologic mechanisms between abdominal obesity and atherosclerotic cardiovascular disease (ASCVD) are known: insulin resistance (IR) and chronic inflammation [13,39]. These are also supported by a study by Lee and Lee [40] who conducted integrative analysis of blood genome-wide transcriptome and disease-gene database. Stanek et al. [41] pinpointed that adverse changes, such as IR and chronic inflammation, trigger and worsen endothelial dysfunction that is the initial stage of ASCVD and other chronic diseases, such as chronic kidney disease and cancers.

Cigarette smoke includes nicotine, carbon monoxide, and oxidant gases, elevating myocardial oxygen demand, reducing myocardial oxygen supply, and driving oxidative stress, which finally results in atherothrombosis, myocardial ischemia, and acute coronary syndrome [42]. Smoking information is used for the binary categorized form (current smoking or not) [43], ternary categorical form (non-, ex-, and current smoker) [44], and continuous variable (pack years) [45]. The majority of ASCVD models implemented the binary categorical variable as predictor [22,23]. Motivated by these studies, we used the binomial distribution of smoking (current smoking or not). The binary distribution of smoking status is mostly obtained from the questionnaire-based method, however, Park et al. [46] reported that Korean women exhibited high rate of false responses for the smoking-related questionnaire, yielding underestimation of the female smoking rate due to certain cultural characteristics. Future study should be required to identify which smoking indices and measurement methods are appropriate for ASCVD or other chronic diseases.

Our study had several limitations. First, the study exploited cross-sectional datasets, so the causal relationship remains unknown. Future work using longitudinal data is required to clarify the etiology. Second, our study analyzed all types of strokes as a result of limited information. Previous studies have analyzed all types of strokes, including ischemic and hemorrhagic strokes, yielding valuable information [47,48]. However, a study concentrating on a subtype of stroke is required to reveal the differences in the risk patterns of ischemic and hemorrhagic strokes. Third, we did not include sufficient sources for defining chronic diseases, such as diabetes and dyslipidemia. Diabetes was defined when fasting plasma glucose ≥ 126 mg/dL or 2 h plasma glucose 200 mg/dL during oral glucose tolerance test (OGTT) or HbA1c ≥ 6.5% in several studies [49]. Other studies implemented the International Classification of Disease guideline (ICD) to define chronic diseases, such as diabetes and hypertension [50]. Numerous samples in KNHANES had missing data for HbA1c and did not have any data for OGTT and ICD codes. Therefore, we depended on the self-reported data (questionnaires) to define the chronic diseases.

## 5. Conclusions

The present study elucidated the association between abdominal obesity and the increased prevalence of stroke in Korean women as well as the non-linear relationships between them in both men and women. As easily accessible and powerful markers for primary prevention for stroke, WC and its adjusted (e.g., waist-hip-ratio and waist-height-ratio) form might be helpful in clinical and public health settings. Our study may motivate future investigations to conduct sophisticated risk analysis by clustering-based or subgroup analyses to reveal several association patterns between WC and stroke.

## Figures and Tables

**Figure 1 ijerph-19-06140-f001:**
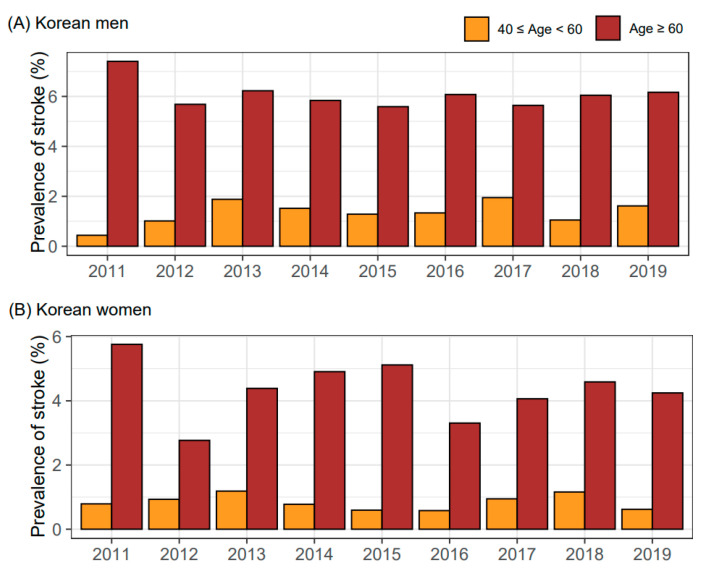
Sex-specific prevalence of stroke according to the 2011–2019 KNHANES datasets. Prevalence was calculated using weighted Korean men (**A**) and women (**B**). Abbreviation: KNHANES, Korea National Health and Nutrition Examination Survey.

**Figure 2 ijerph-19-06140-f002:**
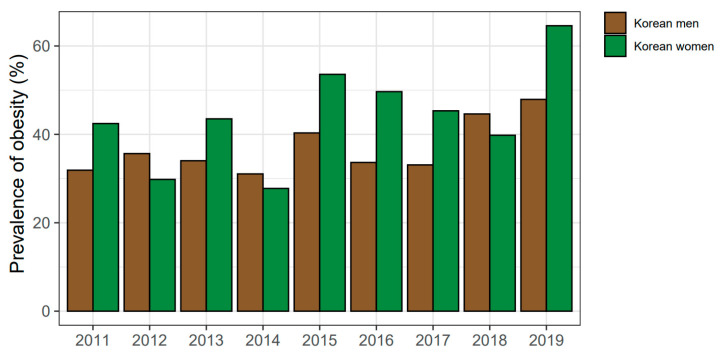
Sex-specific prevalence of abdominal obesity in stroke subjects according to 2011–2019 KNHANES datasets. Prevalence of abdominal obesity (waist circumference ≥90 cm for men and ≥85 cm for women) was calculated by weighted men and women diagnosed with stroke. Abbreviation: KNHANES, Korea National Health and Nutrition Examination Survey.

**Figure 3 ijerph-19-06140-f003:**
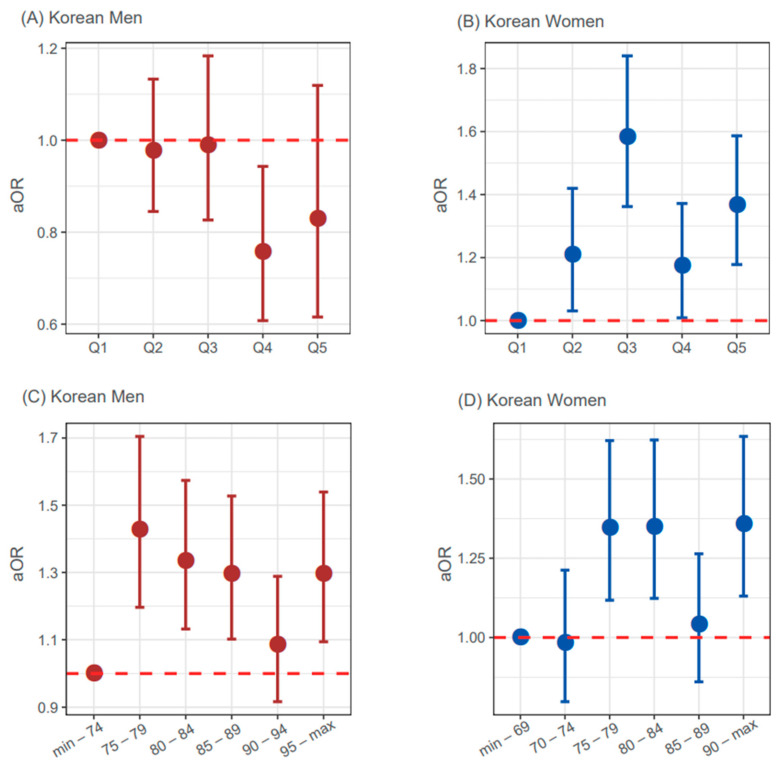
Sex-specific non-linear relationships between waist circumference and the risk for stroke. Adjusted ORs (aORs) were measured using the multivariate logistic regression including age, antihypertensive medication, diabetes, current smoking, systolic blood pressure, total cholesterol, and high-density lipoprotein cholesterol as covariates. The cut-offs of WC were determined based on quintile method (**A**,**B**) and empirical cut-offs (**C**), men: 75, 80, 85, 90, and 95 cm; (**D**), women: 70, 75, 80, 85, and 90 cm).

**Table 1 ijerph-19-06140-t001:** Sex-specific general characteristics according to stroke status.

	Korean Men			Korean Women		
Variable	Non-Stroke	Stroke	*p*-Value	Non-Stroke	Stroke	*p*-Value
Unweighted subjects, n	12,622	507		17,905	456	
Age, years	59.4 ± 0.1	67.6 ± 0.41	<0.001	58.8 ± 0.09	68.4 ± 0.46	<0.001
WC, cm	86.3 ± 0.08	87.7 ± 0.37	<0.001	80.9 ± 0.07	85 ± 0.47	<0.001
<75 (women: 70)	1171 (9.3)	31 (6.1)	0.001	2028 (11.3)	26 (5.7)	<0.001
75 (70)–84 (79)	4232 (33.5)	159 (31.4)		6540 (36.5)	116 (25.4)	
85 (80)–94 (89)	5384 (42.7)	214 (42.2)		6408 (35.8)	178 (39)	
≥95 (90)	1835 (14.5)	103 (20.3)		2929 (16.4)	136 (29.8)	
Abd obesity, n	4137 (32.8)	195 (38.5)	<0.001	5699 (31.8)	213 (46.7)	<0.001
Systolic BP, mmHg	123.2 ± 0.14	126.8 ± 0.8	<0.001	121.4 ± 0.13	130 ± 0.88	<0.001
<130	8634 (68.4)	301 (59.4)	<0.001	12,497 (69.8)	235 (51.5)	<0.001
130–139	2141 (17)	104 (20.5)		2580 (14.4)	92 (20.2)	
140–149	1100 (8.7)	58 (11.4)		1575 (8.8)	76 (16.7)	
≥150	747 (5.9)	44 (8.7)		1253 (7)	53 (11.6)	
Current smoking, n	4050 (32.1)	124 (24.5)	<0.001	676 (3.8)	19 (4.2)	0.758
Antihypertensive medication, n	3727 (29.5)	330 (65.1)	<0.001	5109 (28.5)	313 (68.6)	<0.001
Diabetes, n	2406 (19.1)	187 (36.9)	<0.001	2403 (13.4)	149 (32.7)	<0.001
TC, mg/dL	189 ± 0.33	166.4 ± 1.72	<0.001	197.2 ± 0.28	181.2 ± 1.76	<0.001
HDL-C, mg/dL	46.9 ± 0.1	44.4 ± 0.51	<0.001	52.7 ± 0.09	48.5 ± 0.54	<0.001

Summary statistics for continuous and categorical variables are described as mean ± standard error and number (percent), respectively. Abbreviations: WC, waist circumference; BP, blood pressure; TC, total cholesterol; HDL-C, high-density lipoprotein cholesterol.

**Table 2 ijerph-19-06140-t002:** Association between abdominal obesity and the risk of stroke identified by logistic regression.

Men, WC (Continuous Variable, cm)			Women, WC (Continuous Variable, cm)		
Model	ORs (95% CI)	*p*	Model	ORs (95% CI)	*p*
Univariate model	1.015 (1.011–1.02)	<0.001	Univariate model	1.044 (1.041–1.048)	<0.001
Model 1	1.015 (1.011–1.019)	<0.001	Model 1	1.024 (1.02–1.028)	<0.001
Model 2	0.994 (0.99–0.998)	0.004	Model 2	1.009 (1.005–1.013)	<0.001
Model 3	0.996 (0.992–1.001)	0.089	Model 3	1.009 (1.005–1.014)	<0.001
**Men, Abdominal Obesity (WC ≥ 90)**			**Women, Abdominal Obesity (WC ≥ 85)**		
**Model**	**ORs (95% CI)**	** *p* **	**Model**	**ORs (95% CI)**	** *p* **
Univariate model	1.238 (1.153–1.33)	<0.001	Univariate model	1.892 (1.788–2.002)	<0.001
Model 1	1.187 (1.104–1.276)	<0.001	Model 1	1.242 (1.172–1.317)	<0.001
Model 2	0.886 (0.822–0.955)	0.002	Model 2	0.968 (0.912–1.028)	0.288
Model 3	0.912 (0.846–0.985)	0.018	Model 3	0.968 (0.911–1.028)	0.290

Logistic regression was performed using continuous or binary form of WC and the stroke status as independent and dependent variables, respectively. Model 1: adjusted by age. Model 2: adjusted by Model 1 + antihypertensive medication + diabetes + current smoking + systolic blood pressure. Model 3: adjusted by Model 2 + total cholesterol + high-density lipoprotein cholesterol. Abbreviation: WC, waist circumference; OR, odds ratio; CI, confidence interval.

## Data Availability

The KNHANES is publicly available at https://knhanes.kdca.go.kr/-knhanes/eng/index.do (accessed on 1 December 2021).

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
