# Peer review of "Association between Stroke and Abdominal Obesity in the Middle-Aged and Elderly Korean Population: KNHANES Data from 2011–2019"

_ijerph, 2022, doi:10.3390/ijerph19106140_

Round 1

Reviewer 1 Report

All my comments and sugestions/questions are within the text.

Author Response

We respond comments suggested in other file (pdf).

Reviewer 2 Report

In the manuscript “Association between stroke and abdominal obesity in the Korean population” the authors attempted to analyze the association between stroke and abdominal obesity using nationwide Korea datasets. Although several similar studies have been already performed, the authors performed this study in good number of Korean cohorts. Authors should elaborate result section to make clear sense of what they want to deliver. Here are some of my major concern:

  1. Clear and detailed inclusion and exclusion criteria has not been defined.
  2. Reference for line 43-44 is missing
  3. Recent publication in the research area has not been discussed.
  4. Figure 2 is not clear to me.

Author Response

(The authors gave the same response as above.)

Reviewer 3 Report

  1. Title: Revise as follows: Association between stroke and abdominal obesity in the Korean population: KNHANES data from 2011-2019.
  2. Aim: To clarify the association between stroke and abdominal obesity in Korean population. Instead of "clarify", you may use the word "reveal" or "evaluate".
  3. Results: (a) line 119 - change to "older age" instead of old age. (b) Korean men: WC - only 1.4 cm difference; systolic BP almost within the normal limit. The differences are much larger in Korean women. Instead of mean and SD, it may be more important to find the number (%) who have higher than normal values. I suggest the authors to find the distribution by categories and do a Chi-square test of these two variables. Please follow the categories of WC consistent with the logistic regression data. 
  4. Conclusions: a) Summarize the findings; b) Any future recommendations? c) No new references should be used in conclusions. 

Author Response

(The authors gave the same response as above.)

Reviewer 4 Report

  1. Please provide for how the investigators identify ascertainment methods e.g. Diabetes, current smoker? ICD codes? what codes did you use?
  2. How did the investigators define "abdominal obesity"?
  3. Any data on LDL and Triglyceride that could be incorporated into the analysis. 

Author Response

(The authors gave the same response as above.)

Reviewer 5 Report

The reviewer has a few comments and suggestions for corrections.

In the Abstract, lane 16, the authors write about the method of data analysis – Logistic regression – but do not define at first mention its acronym (LR) that they use later on (lane 19). Only in the Materials and Methods section they clarify this (lane 111).

The same situation happened with an acronym WC in the Abstract (lanes 20, 21, 22) and in the Introduction (lane 40). Only in lane 44 of the Introduction it is defined as waist circumference (WC). That should be clarified at first mention in the Abstract. Similarly, an acronym SBP – systolic blood pressure (lanes 98 and 106).

Lane 50: “in” should be deleted (since people visit something but not in something).

Lane 98: should “current smoker” be replaced with “current smoking” for consistency?

Lane 101: a space should be placed between “… [23].” and “Smoking…”

Lane 102: “…two status” should probably be replaced with “two statuses” since it is plural.

Lanes 201, 203, 204, 206, and 208: if “MetS” is an acronym for “metabolic syndrome” – that should be clarified at first mention.

Lane 235: “concentrating” could be replaced with “focusing” (it’s just an advice). 

Author Response

We respond comments in other file (pdf).

Round 2

Reviewer 1 Report

The authors addressed all questions and doubts and completed the missing information. The work is interesting, I will be happy to use it for quotations in the future. I have no comments.

Reviewer 2 Report

No comment